# MDMX Regulates Transcriptional Activity of p53 and FOXO Proteins to Stimulate Proliferation of Melanoma Cells

**DOI:** 10.3390/cancers14184482

**Published:** 2022-09-15

**Authors:** Renier C. Heijkants, Amina F. A. S. Teunisse, Danielle de Jong, Kseniya Glinkina, Hailiang Mei, Szymon M. Kielbasa, Karoly Szuhai, Aart G. Jochemsen

**Affiliations:** 1Department of Cell and Chemical Biology, Leiden University Medical Center, 2300 RC Leiden, The Netherlands; 2Sequencing Analysis Support Core, Leiden University Medical Center, 2300 RC Leiden, The Netherlands; 3Department of Medical Statistics and Bioinformatics, Leiden University Medical Center, 2300 RC Leiden, The Netherlands

**Keywords:** p53, MDMX, FOXO, MDM2, RFX7, uveal melanoma, cutaneous melanoma, RNA-seq

## Abstract

**Simple Summary:**

We have investigated the transcriptional changes occurring in uveal and cutaneous melanoma cell lines upon depletion of MDMX (aka:MDM4). Computational analyses of the mRNAs/genes affected upon MDMX depletion determined that many were containing a p53-bindingsite, but even more contained a FOX recognition site(s). Since connections between MDM2 and FOXO1 had already been published, we investigated whether indeed a subset of the MDMX-regulated genes are dependent on FOXO1/FOXO3 expression. Indeed, a number of such target genes, i.e., *PIK3IP1, MXD4* and *ZMAT3*, were found to be FOXO target genes in our cell models. Some of these genes were recently identified as indirect p53-target genes, and their expression was found to be regulated by RFX7 transcription factor, which was found activated upon pharmacological activation of p53, e.g., by Nutlin-3. However, a clear involvement of RFX7 in our model could not be established, but an interplay between FOXO and RFX7 factors seems evident.

**Abstract:**

The tumor suppressor protein p53 has an important role in cell-fate determination. In cancer cells, the activity of p53 is frequently repressed by high levels of MDMX and/or MDM2. MDM2 is a ubiquitin ligase whose activity results in ubiquitin- and proteasome-dependent p53 degradation, while MDMX inhibits p53-activated transcription by shielding the p53 transactivation domain. Interestingly, the oncogenic functions of MDMX appear to be more wide-spread than inhibition of p53. The present study aimed to elucidate the MDMX-controlled transcriptome. Therefore, we depleted MDMX with four distinct shRNAs from a high MDMX expressing uveal melanoma cell line and determined the effect on the transcriptome by RNAseq. Biological function analyses indicate the inhibition of the cell cycle regulatory genes and stimulation of cell death activating genes upon MDMX depletion. Although the inhibition of p53 activity clearly contributes to the transcription regulation controlled by MDMX, it appeared that the transcriptional regulation of multiple genes did not only rely on p53 expression. Analysis of gene regulatory networks indicated a role for Forkhead box (FOX) transcription factors. Depletion of FOXO proteins partly prevented the transcriptional changes upon MDMX depletion. Furthermore, depletion of FOXO proteins relatively diminished the growth inhibition upon MDMX knockdown, although the knockdown of the FOXO transcription factors also reduces cell growth. In conclusion, the p53-independent oncogenic functions of MDMX could be partially explained by its regulation of FOXO activity.

## 1. Introduction

The p53 protein is considered to be a master regulator in a cell, mainly due to its central role in cellular stress sensing and its ability to regulate the transcription of a plethora of genes involved in multiple biological processes including cell cycle regulation, apoptosis, metabolism and autophagy [1]. Upon various types of intra- and extracellular stress, p53 is activated and stabilized, resulting in its increased transcriptional activity inducing cell cycle arrest, senescence or apoptosis [1,2]. Due to its central and important function in cell-fate, p53 activity needs to be under tight control. This stringent control of p53 is provided by many proteins including the p53 master-regulators MDM2 and MDMX (aka: MDM4) [3]. This was best demonstrated in vivo where knock-out of *MDM2* or *MDMX* in mice resulted in embryonic lethality, which was shown to be p53-dependent [4,5,6,7,8]. Although both MDM2 and MDMX are crucial for mouse embryonic development, in adult tissue, MDM2 loss is always lethal, whereas depletion of MDMX often results in more mild effects, indicative of the differences in expression between the two proteins [9,10,11,12,13,14,15]. MDM2 is a ubiquitin ligase and has been shown to directly bind with its N-terminal hydrophobic pocket to the N-terminal alpha helix of p53 and subsequently ubiquitinate p53, which consequently is sent for proteasomal degradation [16,17,18,19]. MDMX was discovered as a novel p53 interactor with high sequence homology and great structural similarities with MDM2 and capable of inhibiting p53-mediated transcriptional activation [3,20,21]. Despite the high conservation of the acidic domain and the RING domain, MDMX does not possess detectable ubiquitin ligase activity. However, MDMX can, via its RING finger, form a heterodimer with MDM2, enhancing its ubiquitin ligase activity [22,23]. MDMX directly inhibits transcriptional activation by p53 by tightly binding and thereby shielding its transactivation domain [20,24]. Considering that the levels of MDM2 and MDMX are crucial for the level and/or activity of p53 in a cell, the levels of MDM2 and MDMX are also under stringent control. Upon certain stress, e.g., DNA damage, p53 has to be liberated from its inhibitors MDM2 and MDMX. One proposed mechanism is the degradation of both proteins by increased MDM2-mediated ubiquitination and via decreased de-ubiquitination by USP7 [25,26,27,28]. Additionally, ATM-mediated phosphorylation on MDM2 inhibits its ubiquitin ligase activity towards p53 [25,29]. As a result, the p53 protein is activated and stabilized and can perform its tumor suppressor function. During the recovery phase after an insult, a cell needs to re-constrain p53 activity by MDM2 and MDMX. It has been demonstrated that both *MDM2* and *MDMX* genes are transcriptional targets of p53 providing a negative feedback loop [30,31].

Activity and stability of MDM2 and MDMX is regulated by additional kinases, including AKT, Abl, AXL and CK1 (reviewed in Wade, Li and Wahl, 2013; [32]).

Approximately 50% of all human cancers contain a genetically altered *TP53* gene, either a point mutation or a deletion leading to loss of expression, to render cancer cells resistant to the tumor suppressor function of p53 [33,34]. Despite the high frequency of *TP53* mutations, the other half of the human cancers had to find alternative mechanisms to attenuate p53 signaling [35]. Amplifications of *MDM2* or *MDMX* are frequently found in sarcoma, glioblastoma, retinoblastoma and breast cancer, providing an interesting therapeutic target, i.e., re-activation of wild-type p53 by inhibition of MDM2/MDMX-p53 interaction [36]. Nutlin-3 was the first small molecule compound reported to disrupt the MDM2 and p53 interaction resulting in stabilization of p53, subsequently inducing cell cycle arrest and apoptosis, without activation of the DNA damage response [37]. More recently many other small molecule compounds and peptides have been reported to disturb the MDM2-p53 interaction and some of these have been tested in clinical trials (reviewed in Sanz et al., 2019 [38]). However, since the use of MDM2 inhibitors results in rather strong adverse effects, it might be clinically more interesting to target MDMX in cancer cells [39,40].

Recently, it has been shown that the abundance of full-length MDMX protein can be diminished in tumors by using anti-sense oligonucleotides resulting in a shift from the full length to the short RNA isoform, *MDMX-S*, activating a p53 tumor suppression response [41]. MDMX has been shown to be a relevant therapeutic target in retinoblastoma [42], cutaneous and uveal melanoma [43,44,45] and wild-type *TP53* breast cancer [46]. MDMX depletion induced cell cycle arrest and apoptosis in cancer cells in a partly p53-independent manner [43,44], which could imply that MDMX is not only a valuable therapeutic target in wild-type p53 cancer cells, but also in p53 mutated cells. Indeed, in *TP53*-mutated breast cancer cell lines, containing high levels of MDMX protein, the presence of MDMX was proven to be essential for cell viability and tumor growth [47]. This could be explained, at least partly, by the p53-independent upregulation of CDK-inhibitor p27^KIP1^ upon MDMX depletion [44,47]. Furthermore, it has been reported that high levels of MDMX inhibit the early DNA damage response, independently of p53 and MDM2, resulting in genome instability [48]. An excellent recent review addresses both the regulation, functions and the p53-dependent and -independent functions of MDM2 and MDMX [49].

Even so, the exact mechanisms leading to the p53-independent oncogenic functions of MDMX remain largely unspecified. Here, we set out to elucidate to what extent the transcriptome is controlled by MDMX in a wild-type p53 cell line and how transcription regulation by MDMX might explain both its p53-dependent and -independent functions. Interestingly, we found that depletion of MDMX not only activates p53, but also FOX(O) transcription factors. Furthermore, the growth inhibition upon MDMX depletion appeared to be partly dependent on FOXO proteins.

Therefore, the novelty of this study is that MDMX appears to function as an oncogene by regulation of p53 and FOXO tumor suppressor functions.

## 2. Materials and Methods

### 2.1. Cell Culture and Viability Assays

The UM cell lines MEL202 [50] and 92.1 [51] were cultured in a mixture of RPMI and DMEM-F12 (1:1 ratio), supplemented with 10% fetal calf serum (FCS) and antibiotics.

The MEL202 cell lines was provided by Dr. Bruce Ksander, Dept. Ophthalmology, Harvard University, Boston, USA. The 92.1 cell line was a gift from Dr. Martine Jager, Dept. Ophthalmology, Leiden University Medical Center, Leiden, The Netherlands.

Primary cutaneous melanoma lines MEL 93.05, 94.07, 04.01, 04.04 and 06.24 were cultured in DMEM medium, supplemented with 10% FCS and antibiotics. The primary melanoma cell lines were established from melanoma tissue that was processed according to the Code of Conduct for Responsible Use of human tissues or in the context of study P04.085 that was approved by the Medical Ethics Committee of Leiden University Medical Center and conducted in accordance with the Declaration of Helsinki and after obtaining informed consent.

For growth assays, the cells were seeded in triplicate, in 96-well format and incubated for 3 or 5 days with compounds as indicated. For a 5 day experiment, the cells were replenished with fresh medium/compounds after 3 days. Cell survival was determined via the Cell Titre-Ble Cell Viability assay (Promega, Fitchburg, WI, USA); fluorescence was measured in a microplate reader (Victor, Perkin Elmer, San Jose, CA, USA).

For colony assays, the cells were seeded in triplicate in 12-well plates and were incubated with compounds for 5 days; cells were replenished with fresh medium/compounds after 3 days. Cells were fixed for 10 min in 4% paraformaldehyde. DNA was stained using 30-min incubation with 0.05% crystal violet. After washing and drying, the relative number of cells was quantified by solubilizing the crystal violet in methanol and measuring absorbance at 540 nM using a microplate reader (Syngene HT, Bio-Tek Instruments, Winooski, VT, USA).

Doxycycline was obtained from Sigma-Aldrich (St Louis, MO, USA) and Nutlin-3 from Cayman Chemical (Ann Arbor, MI, USA).

### 2.2. Manipulation of Cell Lines

#### 2.2.1. Establishment of Inducible MDMX Knockdown Cell Lines

Inducible shRNA knockdown lentiviral vectors were constructed as described previously [46,52]. Production of lentivirus stocks by transfections into HEK293T cells essentially as described, but calcium phosphate was replaced with PEI [53]. Virus stocks were quantitated by antigen capture ELISA measuring HIV p24 levels (ZeptoMetrix Corp., New York, NY, USA). Cells were transduced using MOI 2 in medium containing 8 μg/mL polybrene. Target sequences of shRNA constructs to deplete MDMX, FOXOs and shCtrls are shown in Appendix A.

#### 2.2.2. Generation of p53 Knockout MEL202-Derived Cells

MEL202 cells were transduced with puromycin-selectable lentiviral guideRNA expression construct (AA19_pLKO.1-puro.U6.sgRNA.Bvel-stuffer) [54], with targets the sequence: 5′-CCATTGTTCAATATCGTCCG-3′ in exon 4 of the *TP53* gene [55]. After selection on puromycin, Cas9 was temporarily expressed upon transduction with adenoviral Cas9 expression construct (AdV^Δ2^P.Cas9.F^50^ [56]) or a control EGFP encoding adenoviral vector (20 IUs/cell). The EGFP-encoding adenoviral vector used differs from AdV.Δ2.donor^S1/T-TS^ [57] in that it has FRT sites flanking an expression cassette consisting of the human PGK1 gene promoter, the EGFP ORF and the bovine growth hormone polyadenylation signal. Cells were selected for p53 inactivity by continuous presence of 4 µM of Nutlin-3 (Cayman Chemical, Ann Arbor, MI, USA) after which single cell derived clones were established and p53 protein expression was analyzed. To determine the exact mutation, genomic DNA of these clones was isolated, which was used as a template for PCR (Fw primer: 5′-GAGACCTGTGGGAAGCGAAA and Rv-primer: 5′-GCTGCCCTGGTAGGTTTTCT), followed by Sanger sequencing using the forward primer. To investigate chromosomal abnormalities upon CRISPR/Cas9-mediated *TP53*-knockout, karyotyping of the cell lines was performed by COBRA-FISH as described earlier [58,59].

### 2.3. RNA Sequencing

MEL202 cells containing four *MDMX*-targeting doxycycline inducible shRNA constructs and one control shRNA construct were treated for 48 h with 10 ng/mL doxycycline; the cell line containing the control shRNA construct was also mock treated to investigate doxycycline-induced effects. RNA was isolated from three independent biological replicates. RNA was isolated using miRNeasy (Qiagen, Hilden, Germany) and treated with DNAse (Qiagen) according to manufactures protocol. The quality of the samples was determined using a Bioanalyzer 6000 nanochip (Agilent, Santa Clara, CA, USA), followed by ribosomal RNA depletion using Ribo-Zero (Epicentre, Madison, WI, USA), after which, the libraries were constructed as described by the NEBNext Ultra Directional RNA Library Prep Kit for Illumina [NEB E7420S]. Samples were pooled and sequenced on the HiSeq2500, run type paired-end 2 × 50bp + dual index on v4 reagents and flowcells. Read counts were extracted from the BAM files with FeatureCounts (version 1.5.3). For statistical analysis of the read counts, R (version 3.4.2) was used. Read counts were normalized using quantile normalization in the DeSeq package (Version 1.16.1). Differential expression analysis was performed with DeSeq. Gene regulatory network analysis was performed using iRegulon [60]. RNA-seq data set has been deposited to GEO Datasets (NCBI), accession number GSE211802.

Enriched biological processes (GO) were determined using String-DB version 10.5.

### 2.4. RNA Isolation, cDNA Synthesis and Real-Time Quantitative PCR

RNA was isolated using the SV total RNA isolation kit (Promega), after which cDNA was synthesized using the reverse transcriptase reaction mixture as indicated by Promega. qPCR was performed using SYBR green mix (Roche Diagnostics, Indianapolis, IN, USA) in a C1000 touch Thermal Cycler (Bio-Rad laboratories, Hercules, CA, USA). In independent experiments the expression of target genes was determined and normalized to at least two housekeeping genes *CAPNS1* and *SRPR*, and in some experiments also to *RPS11* and *TBP*. Primer sequences are shown in Appendix A.

### 2.5. Western Blot Analysis

After incubation with drugs as indicated, cells were harvested in Laemmli sample buffer. Equal protein amounts were separated using SDS-PAGE and blotted onto polyvinylidene fluoride transfer membranes (Millipore, Darmstadt, Germany). After blocking in TBST (10 mM Tris-HCl pH8.0, 150 mM NaCl, 0.2% Tween 20) containing 10% milk, membranes were incubated with the proper primary antibodies: USP7 (A300-033) and RFX7 (#A303-062A) from Bethyl Laboratories, Montgomery, TX, USA; MDMX (8C6), MDM2 (3G9) and p21 (CP74) from Millipore, Burlington, MA, USA; p53 (DO1), PUMA (G3) and MDM2 (SMP14), all from Santa Cruz Biotechnology, Dallas, TX, USA; MAD2L1 (C2C3, Genetex, Irving, CA, USA); Vinculin (hVIN-1/V9131, Sigma-Aldrich, St Louis, MO, USA; FOXO1 (C29H4) and FOXO3 (75D8); (Cell Signaling Technology, Danvers, Massachusetts, USA, and appropriate HRP-conjugated secondary antibodies (Jackson Laboratories, Bar Harbor, ME, USA). Bands were visualized using chemoluminescence and visualized by exposure to X-ray film or imaged on a Chemidoc machine (Bio-Rad laboratories, Hercules, CA, USA).

### 2.6. Statistical Analysis

Data analysis was performed using GraphPad Prism version 9 for Windows (GraphPad Software, San Diego, CA, USA). The bars present mean ± SEM. The data were analyzed using ANOVA or t-test, the *p*-values < 0.05 were considered to be significant.

## 3. Results

### 3.1. Identification of Genes Transcriptionally Regulated by MDMX

The effect of p53 reactivation on the transcriptome has been studied extensively in previous studies (reviewed by: [61,62]). Here, we studied the regulation of the transcriptome by MDMX. For this purpose, we used a wild-type p53 cell line (MEL202) derived from a primary uveal melanoma, a cancer which rarely has mutated p53 and frequently highly expresses MDMX protein to constrain p53 activity [63]. We have shown before that this cell line is dependent on MDMX expression for proliferation [44]. We generated five MEL202-derived cell lines; one cell line containing a doxycycline-inducible control shRNA expression construct and four cell lines containing distinct doxycycline-inducible *MDMX*-targeting shRNA expression constructs. Efficiency and kinetics of knockdown were tested by incubating the cells with doxycycline for 24, 48 and 72 h and harvesting both RNA and protein for further analysis. These experiments show that the depletion of MDMX was efficient from 24 h onwards with all shRNA constructs used (Figure 1A). Concomitant with MDMX depletion a slight increase in p53 protein levels and p21^CIP1^ protein levels was observed. The kinetics of regulation of gene transcription upon MDMX depletion was further determined by analyzing the mRNA expression of the direct p53 target *CDKN1A* and the indirectly transcriptionally repressed target *MAD2L1* (Figure 1B). An increase in *CDKN1A* expression could already be observed after 24 h and did not increase dramatically in the later time points. Repression of *MAD2L1* expression was only modest at 24 h, but reached a plateau after 48 h.

Based on these results we have performed RNA sequencing upon doxycycline-inducible MDMX knockdown using four different shRNAs and one inducible control shRNA incubated with doxycycline for 48 h. Comprehensive analysis of the data resulted in the identification of 176 genes which were significantly upregulated at least 0.7log2-fold upon MDMX depletion (Appendix A). In addition, the expression of 70 genes was significantly down-regulated with at least −0.7log2-fold upon MDMX depletion, including *MDMX* itself (Appendix A). GO terms pathway analysis showed that many of the upregulated genes promote cell death and apoptosis while the products of the repressed genes are mainly involved in cell cycle regulatory pathways (Appendix A).

To determine via which transcription factor(s) these changes in transcriptome are mediated, we employed the computational method iRegulon, first described to determine the transcriptional effect(s) of Nutlin-3 treatment [60]. According to iRegulon, 66 of the 176 upregulated genes have a p53 binding motive in their promotor region (Appendix A), indicating that, despite the limited log fold changes, at least 37.5% of the upregulated genes could be explained by p53 activation upon MDMX depletion. Interestingly, the majority of the upregulated genes (114 of 176 = 65%) have a Forkhead Box (FOX) motive, recognized by multiple FOX transcription factors. Moreover, genes presenting a p53-binding motive contained also a FOX motive in 89% of the cases. Although iRegulon, in some cases, could not identify either a FOX or p53 motif, a literature search and employing the TargetGeneReg database [64] increased the number of genes with p53 motifs. Indicated in Appendix A are only the analyses of the top 25 upregulated, also including genes containing an RFX7 recognition element (see below).

Analysis of the downregulated genes clearly indicates enrichment for two known repressive transcription (co)regulators, E2F4 and SIN3A. In total, 61 of the 70 (87%) downregulated genes contained one or both repressive transcription regulators binding sites (Appendix A).

We verified the transcriptional regulation of a number of these genes in an independent experiment. Cells were treated for 48 h with doxycycline and RNA expression of some target genes (*KIF23, CENPF, PIK3IP1, PTCHD4*; Figure 2A) was analyzed by real-time qPCR. We included *PIK3IP1* because, from the iRegulon analysis, this gene was not identified as a p53-target gene but as a FOX target, which is confirmed in the literature [65]. From duplicate wells, protein lysates were harvested and investigated and indeed the upregulation of p21^CIP1^ and PUMA and the downregulation of MAD2L1 could be confirmed (Figure 2B). The levels of p53 also slightly increased upon MDMX knockdown, and a small increase in FOXO1 protein could be detected upon MDMX depletion with two of four shRNAs (Figure 2B).

Furthermore, the transcription regulation of selected down- and upregulated ‘MDMX’ target genes were tested upon MDMX knockdown in a second uveal melanoma cell line (92.1), also expressing wild-type p53 and high levels of MDMX. Moreover, in this cell line, upregulation of *CDKN1A, PTCHD4, PIK3IP1, MXD4* and repression of *KIF23* and *EXO1* was observed upon MDMX knockdown (Appendix A).

In addition, we noticed, again, a slight effect on FOXO1 protein levels upon MDMX knockdown in these 92.1 cells (Appendix A). These results indicate that the effects observed in the RNA sequencing experiment are not cell line dependent.

### 3.2. P53-Dependent and -Independent Effects of MDMX Depletion

#### 3.2.1. Nutlin-3 Mediated Regulation of MDMX Target Genes

We and others have previously shown that depletion of MDMX can result in a (partly) p53-independent growth arrest. We, therefore, determined to what extent the observed MDMX-induced effects on the transcriptome are p53-dependent. As an initial step, we investigated whether specific activation of p53 by Nutlin-3 in MEL202 cells would affect the expression of the same genes as observed upon MDMX depletion. As expected, we observed that the mRNA level of a classical p53 target gene, *CDKN1A*, is strongly induced upon Nutlin-3 treatment (Appendix A). Similar effects were observed for *CYFIP2* and *PTCHD4*. *PIK3IP1* expression was also increased upon Nutlin-3 treatment, although the induction was much less compared to the previously mentioned target genes and expression of the supposedly FOX/p53 target gene *MXD4* is only marginally induced. The mRNA levels of all genes downregulated upon MDMX knockdown that were tested (*EXO1, KIF23, MAD2L1* and *MCM10*) were also repressed upon incubation with Nutlin-3 (Appendix A). To exclude a putative cell line dependency of the regulation of these genes by p53, we investigated the effect of Nutlin-3 treatment in 92.1 uveal melanoma and in three cutaneous melanoma cell lines, of which MEL 94.07 contains a p53-inactivating mutation. Nutlin-3 treatment of the uveal melanoma cell line 92.1 showed transcriptional regulation of these genes similar to the results in MEL202 with the exception of *MXD4,* which is not affected (Appendix A). In the cutaneous melanoma cell lines, MEL 04.01 and MEL 06.24, Nutlin-3 treatment increased significantly *CDKN1A, CYFIP2* and *PTCHD4* expression, whereas *PIK3IP1* and *MXD4* levels do not or hardly change (Appendix A). It can be noted that, although *CDKN1A, CYFIP2* and *PTCHD4* responded to Nutlin-3 treatment in every cell line, the increase varies significantly per cell line. From these results, we conclude that from these tested genes upregulated upon MDMX knockdown, *CDKN1A, CYFIP2* and *PTCHD4* are *bona fide* p53 target genes, but the mRNA expression of *PIK3IP1* and *MXD4* are not consistently increased upon p53 activation in our tested cell lines. Nutlin-3 treatment of all cutaneous melanoma lines resulted in a strong decrease in mRNA levels of the tested genes downregulated upon MDMX depletion (Appendix A). Cutaneous melanoma cell line MEL 94.07 has an inactivating mutation in *TP53* gene, which indeed coincides with no observed changes in the mRNA levels of the analyzed genes upon Nutlin-3 treatment.

#### 3.2.2. Effects of MDMX-Knockdown in p53-Depleted Cells

As a second step to study the p53-dependency of the changes in the transcriptome upon MDMX depletion, we created MEL202 cells deficient of p53 expression using CRISPR/CAS9 technology. We selected two clones which fully lack any detectable p53 protein as a consequence of a one-nucleotide insertion (Adenine after nt 143 position of the coding sequence) resulting in the mutation of ASP48 to GLU48 and introducing an alternative reading frame resulting in an early translation stop after amino acid position 50 (Appendix A). As a control, we used cells expressing the same guide-RNA, but without exposure to Cas9 expression. The lack of p53 protein renders the cells, as expected, completely resistant for Nutlin-3 induced growth arrest (Appendix A). The p53-KO cells did not show any detectable aberrations in their karyotype at the chromosomal level compared with the parental cell line as determined by COBRA FISH (Appendix A). From these MEL202/CR-Ctrl and MEL202/CRp53 cells doxycycline-inducible shMDMX or shCtrl cell lines were generated and treated with vehicle or with doxycycline and mRNA expression of selected genes was analyzed (Figure 2D). Results show that the increased expression of *CDKN1A*, *PTCHD4* and *CYFIP2* upon MDMX depletion is fully p53-dependent (Figure 2C and data not shown). The basal levels of these genes are also strongly reduced in the p53-KO cells, showing some basal p53 activity in the parental cells. Interestingly, expression of the FOX target gene(s) *PIK3IP1* and *MXD4* are still increased upon MDMX depletion in the MEL202 p53-KO cells, although the induction is somewhat less (Figure 2C).

This result fits with the observation that also in the p53-KO cells the FOXO1 protein levels are slightly increased upon MDMX depletion (Figure 2D).

We again analyzed 92.1 cells to confirm these data, albeit that p53 was stably knocked down with shRNA, as described before [66]. Moreover, in these cells, the expression of *CDKN1A* and *PTCHD4* is p53-dependently upregulated upon MDMX depletion, but increased expression of *PIK3IP1* and *MXD4* is p53-independent (Appendix A). Since *PIK3IP1* was identified as a FOX(O) target gene, protein expression of FOXO1 was also investigated from cells treated in the same way. Indeed, upregulation of FOXO1 protein levels in 92.1 cells upon MDMX depletion was found to be p53-independent (Appendix A). These data show that not all genes transcriptionally upregulated upon MDMX depletion are direct p53 target genes in these cell lines, possibly explaining the p53-independent growth stimulatory functions of MDMX.

Similarly, we found that in both MEL202 and 92.1 cell lines, the transcriptional repression of genes upon MDMX knockdown in many cases is partially, but not fully dependent on p53 expression (Figure 2C and Appendix A).

To verify the effect of MDMX depletion on transcriptional regulation of the above-mentioned target genes in non-uveal melanoma cell lines, we used the cutaneous melanoma cell line MEL 93.05. This cell line also expresses wild-type p53 and high levels of MDMX.

We first made derivatives either stably expressing shRNA targeting p53, or shCtrl, and subsequently introduced the inducible shCtrl, shMDMX#1 and shMDMX#2 expression cassettes. These six cell lines were treated with doxycycline or solvent, in the presence or absence of Nutlin-3 for 72 h, after which protein and RNA were harvested for analysis.

MDMX is efficiently depleted upon induction of the shMDMX shRNAs, and Nutlin-3 treatment also slightly reduced the MDMX levels (Appendix A). The depletion of p53 in the shp53 cells is efficient upon doxycycline treatment, but Nutlin-3 treatment still increases p53 protein to significant levels.

Analysis of p53 target genes *CDKN1A, MDM2* and *PTCHD4* indicates an increase upon MDMX depletion in the shCtrl cells, which is abrogated in the shp53 cells (Appendix A)

Nutlin3-treatment strongly increases the mRNA levels of these genes, and the combination of MDMX-depletion and Nutlin-3 increases the expression even further in the shCtrl cells. In the shp53 cells, Nutlin-3 can still transcriptionally activate these genes, but to much lower levels than in control cells. *PIK3IP1* and *MXD4* transcription is increased upon MDMX depletion, in a largely p53-independent manner. In these cells, Nutlin-3 treatment very well induced *PIK3IP1* expression and also slightly increased *MXD4* mRNA levels. The combination of MDMX depletion and Nutlin-3 treatment increased the expression of these genes further, especially of *PIK3IP1*, in a partly p53-dependent manner.

These data obtained in the cutaneous melanoma cell line MEL 93.05 largely confirm the results described above in the MEL202 and 92.1 uveal melanoma cell lines.

### 3.3. FOXO-Dependent and -Independent Effects upon MDMX Depletion

#### 3.3.1. Effects of FOXO Depletion upon MDMX Knockdown: Transcriptional Consequences

To investigate in more detail the importance of FOXO transcription factors in the changes in the transcriptome upon MDMX depletion and to what extent p53 is involved in the FOXO effects, an extra set of cell lines was generated.

First, MEL202/CR-Ctrl and MEL202/CRp53 cells were transduced with lentiviruses either expressing a doxycycline-inducible control shRNA (Luciferase-targeting; i-shCtrl) or a doxycycline-inducible shRNA targeting *FOXO1/FOXO3* mRNAs. These lentiviruses have recently been described by Hornsveld et al. (2018) [67].

Subsequently, these four cell lines were transduced with the above-described lentiviruses containing the doxycycline-inducible i-shMDMX#1 or i-shMDMX#2 cassette or i-shCtrl.

Stable cell lines were treated with doxycycline for 72 h, after which the cells were harvested and protein and RNA extracted. In Figure 3A, the protein expression of FOXO1, FOXO3 and MDMX is investigated to verify the depletion by the respective shRNAs.

Subsequently, the mRNA expression of a number of genes was investigated (Figure 3B), including ‘classical’ p53 target genes such as *CDKN1A* and *MDM2*, and the-putative-FOX(O) targets *PIK3IP1* and *MXD4*.

As can be seen, in the MEL202/CR-Ctrl cells the MDMX knockdown results increased mRNA levels of *MDM2*, *CDKN1A* and *PTCHD4*, which are almost completely lost in the MEL202/CR-p53 cells. Depletion of FOXO proteins does not affect levels of *MDM2*, but slightly increases *CDKN1A* and *PTCHD4* levels. In addition, depletion of FOXOs reduces the increase in *MDM2* and *PTCHD4* levels upon MDMX knockdown a little, but does not affect the increase in *CDKN1A*. We included in our analyses of these cells the *ZMAT3* gene, previously described as a p53 target gene [68] and according to iRegulon also a FOX target gene. MDMX depletion increases *PIK3IP1, MXD4* and *ZMAT3* expression in CR-Ctrl/i-shCtrl cells, which is completely prevented in CR-Ctrl/i-shF1F3 cells. Moreover, in CRp53/i-shCtrl cells, MDMX depletion still induces *PIK3IP1* and *MXD4* expression, but not *ZMAT3* expression. The simultaneous depletion of the FOXOs completely prevents the induction of *PIK3IP1, MXD4* and *ZMAT3*. Furthermore, depletion of FOXOs reduces basal levels of these genes, especially of *PIK3IP1* and *ZMAT3*. These results indicate that the increase in mRNA levels of a number of genes upon MDMX knockdown is mainly dependent on FOXO expression, while other(s), i.e., *ZMAT3,* need both the expression of p53 and FOXO to be regulated by MDMX depletion. The same results were obtained in independent biological experiments.

To verify these results in another cell line, we again used the 92.1 uveal melanoma cell line.

These cells were first transduced with lentiviruses expressing either the doxycycline-inducible control shRNA or the doxycycline-inducible FOXO1/FOXO3 shRNA. Subsequently, these cells were transduced with either i-shCtrl, i-shMDMX#1 or i-shMDMX#2.

Knockdown of FOXOs and MDMX was checked by analyzing protein lysates harvested after 72 h of treatment with doxycycline or vehicle (Appendix A).

Despite the limited knockdown of MDMX knockdown by i-shMDMX#2, in the i-shF1F3 background, RNA was harvested from duplicate plates and mRNA expression of selected genes was analyzed (Appendix A).

Depletion of MDMX increases mRNA expression levels of *MDM2*, *CDKN1A* and *PTCHD4*, in a partly FOXO-dependent manner. *PIK3IP1, MXD4* and *ZMAT3* expression are also well induced upon MDMX depletion in 92.1 cells. The increase in *PIK3IP1* and *ZMAT3* is essentially abrogated upon simultaneous depletion of FOXOs, as seen in MEL202 cells. Surprisingly, in 92.1 cells, the depletion of FOXO proteins results in a slight increase in *MXD4* expression, but that expression is not further increased upon simultaneous MDMX depletion, still indicating that the enhanced expression upon MDMX depletion needs the presence of FOXO proteins.

The i-shMDMX#2 cell line shows hardly any induction of the p53-target genes in the i-shF1F3 background, but that can be explained by the inefficient knockdown of MDMX in this cell line.

#### 3.3.2. Effects of FOXO Depletion upon MDMX Knockdown: Growth Consequences

The same sets of MEL202- and 92.1-derived cell lines were used to investigate whether the growth inhibition upon MDMX knockdown is, indeed, partly p53-independent, as we mentioned before, and whether the regulation of FOXOs is involved in this growth inhibition.

Therefore, we determined the relative survival after 5 days of doxycycline treatment of the MEL202-derived cell lines. In Figure 4A, one representative of such an experiment is shown. Several conclusions can be drawn from these experiments.

First, as shown before [44], growth inhibition upon MDMX knockdown in MEL202 cells is partly p53-independent (relative survival for i-shMDMX#1 and i-shMDMX#2 is 0.271 and 0.248 in CR-Ctrl and 0.531 and 0.434 in CR-p53 cells).

Secondly, knockdown of the FOXOs also inhibits the growth of MEL202 cells, again partly p53-dependent (relative survival 0.286 and 0.489, respectively), indicating a proliferative function of FOXO proteins in these cells.

Simultaneous depletion of MDMX and FOXO proteins reduces the relative survival of the cells more than either single knockdown. However, if the effect of MDMX knockdown is calculated relative to either the i-shCtrl + Doxy or the i-shF1F3 + Doxy, it can be seen that in the FOXO-depleted situation the knockdown of MDMX has less of a growth inhibitory effect (Figure 4B).

In the CR-Ctrl background, relative survival is 0.288 and 0.264 for i-shMDMX#1 and i-shMDMX#2, respectively, relative to i-shCtrl/Doxy, and 0.425 and 0.606 for i-shMDMX#1 and i-shMDMX#2, respectively, relative to i-shF1F3/Doxy. In the CR-p53 background, relative survival is 0.524 and 0.428 for i-shMDMX#1 and i-shMDMX#2, respectively, relative to i-shCtrl/Doxy, and 0.651 and 0.635 for i-shMDMX#1 and i-shMDMX#2, respectively, relative to i-shF1F3/Doxy. These results suggest that MDMX stimulates the growth of uveal melanoma cell lines by attenuating both p53 and FOXO activity.

The depletion of FOXO proteins has a strong effect on the relative survival of the MEL202 cells after 5 days of doxycycline, especially in the CR-Ctrl cells; therefore, we decided to perform the same experiments again with 3 days of doxycycline treatment (Figure 4C). As expected, after 3 days of knockdown of MDMX or FOXOs, the growth inhibition is much less compared to 5 days knockdown, and, especially, the MDMX knockdown is clearly partly p53-dependent. The effect of FOXO knockdown is now almost similar in the CR-Ctrl and CR-p53 cells. We have noticed more often that the difference between CR-Ctrl and CR-p53 in effect of FOXO knockdown takes relatively a long time, suggesting an indirect p53-dependent effect.

The depletion of FOXO in CR-Ctrl cells reduces relative survival to 71%; additional knockdown of MDMX reduces this further to 44% and 59% (i-shMDMX#1, i-shMDMX#2, respectively). On the other hand, in i-shCtrl cells MDMX depletion reduces the survival from 92% to 46% and 47%. Again, this means that the effect of MDMX depletion is stronger in i-shCtrl cells than in i-shFOXO cells, as shown in Figure 4D. In the CR-p53 cells, the differences are less, and actually are not present upon depletion of MDMX with i-shMDMX#1, but are still clear in the i-shMDMX#2 cells.

To show that the effects of MDMX knockdown and FOXO knockdown is not cell line dependent, we used again the 92.1 cell lines i-shCtrl and i-shF1F3, containing i-shCtrl, i-shMDMX#1 and i-shMDMX#2. The cells were treated for 72 h with doxycycline and relative survival was determined. In the control cells, the survival upon MDMX knockdown by both shRNAs is about 35%. FOXO knockdown reduces survival to 85%. Compared to MEL202 cells, the 92.1 cells appear a bit more sensitive to MDMX depletion but slightly less sensitive to FOXO depletion. The combined depletion of MDMX and FOXO proteins reduces relative survival to about 52% and 72%, so even less effect than only MDMX depletion (Appendix A). As seen in MEL202 cells, the effect of MDMX knockdown in the FOXO-depleted cells on the growth of the cells is reduced, as can be clearly seen in Appendix A.

Very similar observations were made in two additional independent biological replicate experiments in these 92.1 cells.

These results obtained in both MEL202 and 92.1 uveal melanoma cells suggest that indeed the oncogenic function of MDMX is not only the attenuation of p53 tumor suppressor activity but also by diminution of a tumor suppressor function of FOXO transcription factors.

### 3.4. Regulation of RFX7 and RFX7-Target Genes upon MDMX Depletion

Lastly, during the course of these studies it has been reported that the p53-target gene encoding the transcription factor RFX7 is involved in the p53-dependent upregulation of mRNA levels of genes in which no direct p53-binding site in the promoter region could be found [69].

These studies, although no clear upregulation of *RFX7* at the mRNA level could be found, indicated that the expression of a number of genes, including *PIK3IP1* and *MXD4*, which we identified as FOX(O) target genes, is indirectly increased upon p53 activation via stimulating RFX7 activity. Furthermore, some other genes relatively high up in our list of upregulated genes upon MDMX depletion, including *DDIT4, TP53INP1* and *PNRC1,* are reported as RFX7-target genes [69]. Based on these results, the website regarding p53-target genes was updated to TargetGeneReg 2.0 [70], including information on the RFX7 target genes. Therefore, we decided to investigate a possible involvement of RFX7 in our cell lines upon MDMX depletion.

As a first step, the mRNA expression of a number of the above-mentioned genes was investigated on the same samples as shown in Figure 3B, so MDMX-depletion in MEL202/CR-Ctrl and/CR-p53 cells, either in presence of absence of simultaneous FOXO-depletion (Figure 5A). We did not find upregulation of *RFX7* mRNA levels upon depletion of MDMX, or in CR-Ctrl nor in CR-p53 cells, but that is not necessarily in conflict with the published results on Nutlin-3 treated cells [69]. Interestingly, some minor, but significant, upregulation (~25–35%) of *RFX7* could be found upon reducing FOXO levels, in a p53-independent manner. The expression of *RFX5* mRNA, identified as an RFX7 target gene, is slightly downregulated upon MDMX depletion, in a p53-dependent manner. Depletion of FOXO results in downregulation of *RFX5* mRNA levels (40–50%) in a p53-independent manner. These results do suggest some cross-talk between RFX5/RFX7 and FOXO transcription factors.

As published [69], activation of p53, in this case by MDMX-depletion, resulted in an increase in *PNRC1* mRNA levels, which is partly p53-dependent. Interestingly, depletion of FOXO results in a slight increase in *PNRC1*, which is enhanced by simultaneous MDMX depletion in a partly p53-dependent manner, which could be the result of increased *RFX7* expression upon FOXO depletion. Most strikingly is the regulation of *PDCD4* mRNA levels; we find no increase at all upon MDMX depletion, but a slight downregulation in a p53-dependent manner. Reducing FOXO levels more strongly downregulates *PDCD4* expression levels (50–60%), essentially in a p53-independent manner. These *PDCD4* results are difficult to reconcile with the earlier mentioned publication [69], indicating that *PDCD4* is an -indirectly-p53-upregulated gene, although we cannot exclude cell line specificity.

Therefore, we performed the same analyses in the 92.1 cells on the same samples shown in Appendix A. It was found that *RFX7* mRNA levels are not influenced by either MDMX or FOXO depletion in these cells (Appendix A). *RFX5* expression is again slightly downregulated upon FOXO depletion and no effect of simultaneous MDMX depletion. As in MEL202, *PNRC1* mRNA levels are induced upon MDMX depletion, which is somewhat reduced in the combined FOXO/MDMX depleted samples, suggesting an involvement of FOXOs. *PDCD4* is also not induced upon MDMX depletion in 92.1 cells, but knockdown of FOXO again reduced *PDCD4* levels. These results largely confirm the data obtained in the MEL202 cells.

Since in the previous publication the induction of these RFX7 target genes was not investigated upon MDMX depletion, but only via pharmacological activation of p53, including Nutlin-3, the expression of these genes was analyzed in the MEL 93.05 samples in which the cells were also treated with Nutlin-3 (Appendix A).

In the 93.05/shCtrl cells the *RFX7* expression is slightly increased upon MDMX knockdown, by Nutlin-3, and slightly more by the combination. In the 93.05/shp53 cells only the depletion of MDMX increases slightly the *RFX7* mRNA levels, and no effect of Nutlin-3 anymore.

*RFX5* levels are not consistently affected and are not reflecting the effects on *RFX7.* As in the uveal melanoma cell lines expression of *PNRC1* is induced upon MDMX depletion, in a largely p53-independent manner. Nutlin-3 also increases *PNRC1* levels, but that is very much p53-dependent, illustrating once more that reduction of MDMX levels in resulting in distinct effects on the transcriptome than activation of p53.

In these 93.05 cells just the doxycycline treatment in shCtrl/i-shCtrl and shp53/i-shCtrl does increase the expression levels of *PDCD4* a little, making further conclusions difficult. However, it is quite clear that this induction is abrogated upon MDMX depletion in the shCtrl cells, and is changed into a reduction upon MDMX knockdown in the shp53 cells.

Therefore, in these cells the *PDCD4* mRNA levels do not follow the RFX7 expression.

Lastly, we treated three cutaneous melanoma cell lines with Nutlin-3 for 24 h and investigated the mRNA expression of *RFX7, RFX5, PNRC1, PDCD4* and *CDKN1A* as positive control (Appendix A). In all three tested cell lines, *RFX7* expression is increased upon Nutlin-3 treatment 1.6 to 2-fold. Even so, mRNA levels of *RFX5* and *PDCD4* are decreased by Nutlin-3 treatment, while *PNRC1* levels are upregulated similarly to *RFX7.* The increases in expression levels of *RFX7* and *PDCD4* are significant, but very minimal compared to the classical target gene *CDKN1A*.

Further involvement of RFX7 in the response to MDMX depletion was investigated by analyzing RFX7 protein levels. In the publication by Coronel et al. (2021), the authors showed that in some cases not so much the levels of RFX7 protein were increased upon Nutlin-3a treatment, but they noticed a shift towards a somewhat faster migrating protein species, an effect apparently p53-dependent.

We initially analyzed the same samples as were shown in Figure 3A, which shows the FOXO and MDMX depletion in MEL202-derived cell lines. RFX7 is a large protein of about 230 kDa and with our antibody, also used by Coronel et al. (2021), we noticed several background bands. Even so, in Figure 5B, the knockdown of MDMX in MEL202/CR-Ctrl cells indeed seems to result in a partial shift of RFX7 protein, best seen in i-shMDMX#1 samples (see arrows). The size of these bands fits with the expected size of RFX7 protein. This effect on the protein size is not noticed in the MEL202/CR-p53 cells, consistent with the data of Coronel et al. (2021) showing a p53-dependency of this effect on protein mobility.

Interestingly, depletion of FOXO increases RFX7 protein levels, partly p53-dependent. This observation could explain the increase in *PNRC1* mRNA levels, but is not in line with the downregulation of *RFX5* and *PDCD4* mRNA levels upon FOXO depletion, suggesting a strong context dependency of the effects of FOXO-depletion and/or RFX7 protein activity.

These results strongly suggest an interplay between FOXO transcription factor(s) and RFX7, to some extent orchestrated in a p53-dependent manner in MEL202 cells.

The effect on the RFX7 protein levels is consistent in a number of independent experiments in MEL202 cells. In the other uveal melanoma cell line, 92.1, the effect is less clear, but a slightly faster migrating band is appearing just below the main RFX7 band upon MDMX-depletion (Figure 5C).

These experiments show that depletion of MDMX affects the migration of RFX7 protein, but we do not see a consistent upregulation of RFX7 target genes like *RFX5* and *PDCD4*. In our cell systems the upregulation of ‘RFX7’ target genes *PIK3IP1* and *MXD4* upon MDMX depletion is almost completely abrogated upon FOXO knockdown. Together these data could suggest that RFX7 and FOXO cooperate in the regulation of these genes, although our observation that RFX7 protein increases upon FOXO depletion is difficult to reconcile with that model.

## 4. Discussion

Our studies presented here make it evident that MDMX is affecting the transcription of genes which are implicated in cell cycle regulation and apoptosis. Further analysis of the downregulated MDMX target genes indicates that E2F4- and SIN3A-mediated transcription repression is involved. To determine whether SIN3A or E2F4 is essential for the repression of these target genes will be essentially impossible; firstly, because it has been shown that in SIN3A knock-out cells E2F target genes are regulated [71] and, secondly because previous studies actually showed that SIN3A forms a repressive complex with E2F4 [72,73]. Most likely, a large repressive complex is formed, containing multiple transcription repressive factors like HDACs, to ensure proper target gene repression.

Based on a large meta-analysis performed by Fisher and colleagues, our results suggest that depletion of MDMX liberates p53, resulting in increased levels of p21 protein which in turn leads to the activation of the DP, RB-like, E2F4 and Muvb (DREAM) complex, resulting in the repression of target genes [64,74,75]. Of the 210 essential regulators of G2 phase and mitosis assigned to be regulated by the p53-p21-DREAM axis [74], we identify 23 genes down-regulated upon MDMX depletion. The identification of the p53-p21-E2F4/SIN3A axis for 87% of the repressed genes seems to indicate that MDMX controls the transcriptional activity of p53 to control cell cycle progression. However, MDMX clearly has (wt)-p53 independent oncogenic functions [43,44,47,48]. Such a p53-independent biological function could correlate with our observation that in distinct cell lines the repression of target genes upon MDMX depletion is not fully p53-dependent. It suggests that MDMX depletion is triggering E2F4/SIN3A-mediated target gene repression in part independent of p53.

The list of upregulated genes upon MDMX knockdown contains 66 genes potentially regulated directly by p53 of which multiple genes are among the most commonly found p53-activated genes, namely, *CDKN1A*, *TP53INP1*, *TSPAN11*, *EDA2R* and *CYFIP2* [61], indicating that p53 transactivation is repressed by MDMX in our uveal melanoma cells as expected. Surprisingly, although the upregulation of most verified genes identified as p53 targets upon MDMX depletion is highly dependent on p53 expression, the regulation of at least *PIK3IP1, MXD4* and *PNRC1* is not, or only partially. All three genes were identified as ‘p53-responsive’ in previous studies, but since a clear p53 binding near the promoter region could not be found, they were not assigned as true p53 target gene status [64]. Also in our hands, the responsiveness of these genes to p53 activation upon Nutlin-3 treatment is rather weak. Interestingly, all three genes contain, like 65% of all the up-regulated genes upon MDMX depletion, Forkhead box (FOX) transcription factor DNA binding sites according to iRegulon. *PIK3IP1* has previously been demonstrated to be a direct FOXO3 target [65]. We could verify that FOXO1 protein levels are affected upon MDMX depletion in a p53-independent manner, possibly enhancing the activity of FOXO1, although the effect varied somewhat between the different MDMX targeting shRNA. Most importantly, depletion of FOXO1/FOXO3 prevented the upregulation of *PIK3IP1* and *MXD4* and it could be hypothesized that MDMX could also bind and inhibit the activity of FOXO transcription factors. However, we have tried immunoprecipitations multiple times with both FOXO1 and MDMX antibodies in at least two different cell lines but no convincing interaction could be determined.

We could conclude from our double-inducible FOXO/MDMX knockdown cell lines that the growth inhibition upon MDMX knockdown is partly dependent on the expression of FOXO proteins. However, FOXO depletion inhibits the growth of our tested UM cell lines, also indicating a proliferative function of these proteins, as has been reported for breast cancer cell lines [67]. Although out of the scope of this study, it would be interesting to investigate whether the here-identified FOXO target genes up-regulated upon MDMX depletion play a role in the growth inhibition. PIK3IP1 and PNRC1 have been shown to function as tumor suppressor proteins [76,77], so it could have an antiproliferative function in our cell models as well. Similarly, MXD4 is an antagonist of cMYC by competing with MAX for dimerization and can prevent cMYC-mediated transformation [78].

Recently, an important function for the transcription factor RFX7 was reported explaining the transcriptional upregulation of genes upon p53-activation but containing no p53 binding site, including some of the genes found in our study like *PIK3IP1*, *MXD4* and *PNRC1* [69]. We noticed some effect of MDMX depletion on RFX7 protein, but our results do not yet indicate an important role for RFX7 activation for the transcriptional activation of these genes. Cooperation between RFX7 and FOXO protein(s) can be suggested, also because in the study on RFX7 as p53-target it was found that depletion of RFX7 leads to reduction in the expression of genes with a FOX(O) motif. Preliminary experiments aimed to detect protein-protein interaction of RFX7 with FOXO1 were not successful.

## 5. Conclusions

In conclusion, our data presented here suggest that high MDMX protein levels stimulate the proliferation of cancer cells by inhibiting the activity of p53 and FOXO transcription factor(s). Targeting MDMX as a treatment for cancer should, therefore, be focused on the depletion of MDMX protein levels and not only on the interference with p53-MDMX interaction.

## Figures and Tables

**Figure 1 cancers-14-04482-f001:**
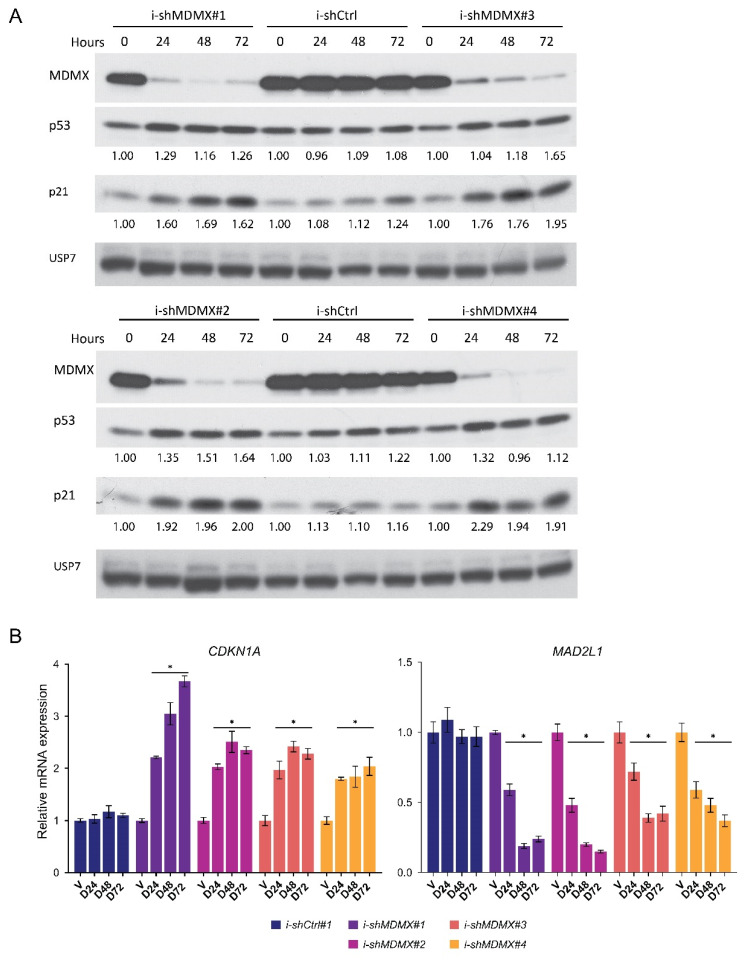
Kinetics of p53 activation upon MDMX depletion in MEL202 cells. (**A**) Protein expression analysis of i-shCtrl and i-shMDMX MEL202 cells harvested after different incubation periods with doxycycline (0, 24, 48 and 72 h). The cells containing the distinct *MDMX* targeting shRNA constructs (#1, 2, 3 and 4) show a clear reduction of MDMX protein upon doxycycline treatment (10 ng/mL). Simultaneously with MDMX depletion, p53 levels slightly increase and p21^CIP1^ levels rise, mostly at the later time points. (**B**) Normalized, relative mRNA expression of *CDKN1A* and *MAD2L1* in i-shCtrl and i-shMDMX MEL202 cells harvested after different incubation periods with doxycycline (D; 10 ng/mL) or Vehicle (V). Expression of *CDKN1A* is markedly increased upon MDMX depletion already after 24 h and only slightly increases at later time points. Repression of *MAD2L1* upon MDMX depletion takes approximately 48 h before reaching a plateau. Significant alterations (*p* < 0.05) in expression levels are indicated with *. Original blots see Appendix A.

**Figure 2 cancers-14-04482-f002:**
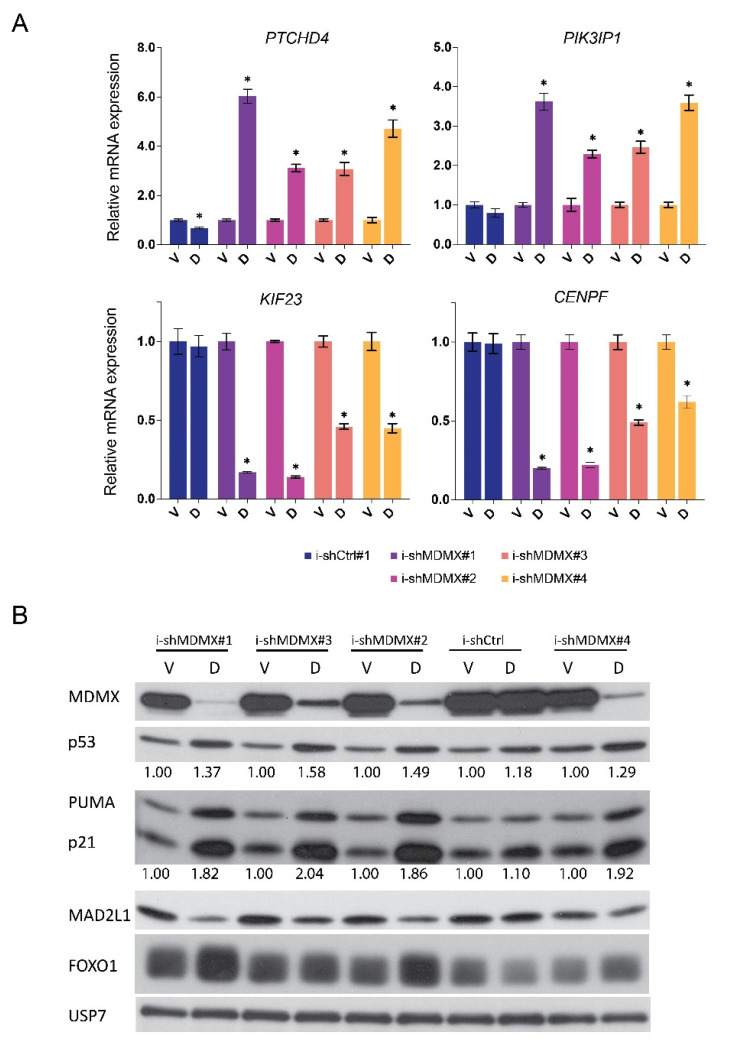
Verification of MDMX target genes. (**A**) Relative, normalized mRNA expression of *CENPF*, *KIF23*, *PIK3IP1* and *PTCHD4* genes after 48 h of doxycycline (D 10 ng/mL) or Vehicle (V) treatment of indicated cell lines. *CENPF* and *KIF23* are downregulated and *PIK3IP1* and *PTCHD4* are upregulated upon MDMX knockdown. Significant alterations (*p* < 0.05) in expression levels are indicated with *. (**B**) Analysis of protein expression after 48 h doxycycline treatment shows a consistent repression of MAD2L1 level and an increase in p53, FOXO1, p21 and PUMA levels upon MDMX depletion. (**C**,**D**) MEL202/CR-Ctrl and MEL202/CR-p53 cells were depleted for MDMX (48 h doxycycline (D), 10 ng/mL) or control-treated (V) after which RNA and protein lysates were harvested and analysed by qRT-PCR (**C**) and Western blotting (**D**). Significant alterations (*p* < 0.05) in expression levels are indicated with *. Original blots see Appendix A.

**Figure 3 cancers-14-04482-f003:**
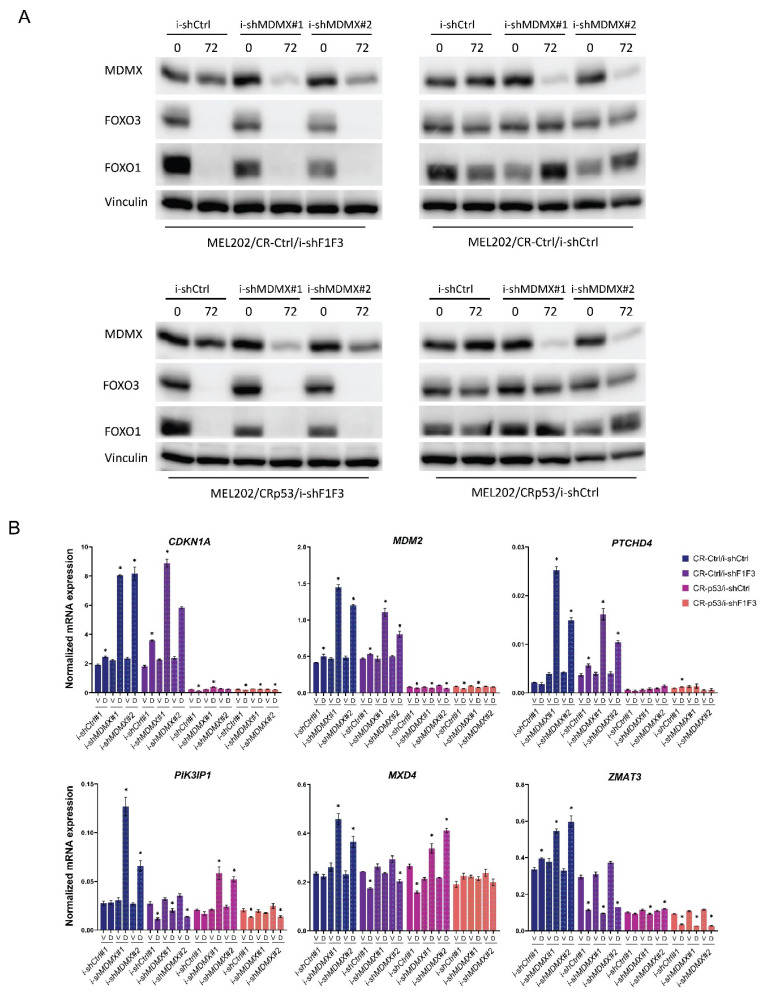
Expression of some MDMX target genes is FOXO-dependent. Analyses of MEL202/CR-Ctrl and MEL202/CRp53 cell lines, with and without 2 distinct i-shMDMX shRNAs, with and without i-shFOXO, plus and minus doxycycline treatment. (**A**) Indicated cell lines were treated for 72 h with 10 ng/mL doxycycline after which protein lysates were harvested and expression of designated proteins was analyzed to assess the efficacy of depletion. (**B**) The various cell lines were treated for 72 h with 10 ng/mL doxycycline (D) or Vehicle (V) after which RNA was harvested and the expression of indicated genes was analyzed by real-time qPCR. Significant alterations (*p* < 0.05) in expression levels are indicated with *. Original blots see Appendix A.

**Figure 4 cancers-14-04482-f004:**
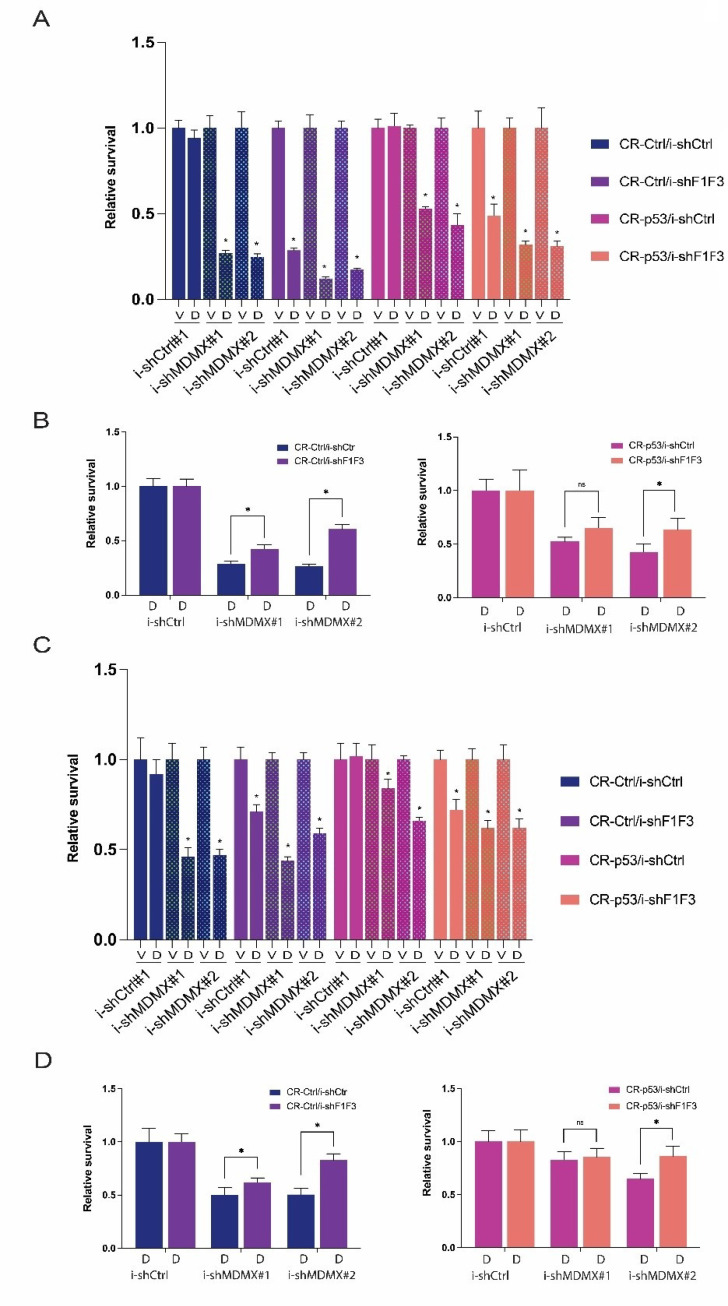
MDMX stimulates growth of UM cell lines MEL202 via attenuation of p53 and FOXO activity. (**A**) Indicated MEL202-derived cell lines (CR-Ctrl and CR-p53; i-shCtrl or i-shFOXO) either containing an i-shCtrl or two distinct i-shMDMX shRNA constructs, were treated with doxycycline (D; 10 ng/mL) for 5 days. Relative survival of the cell lines, each normalized to Vehicle (V)-treated samples. Significant alterations (*p* < 0.05) in cell survival are indicated with *. (**B**) Relative survival of the indicated cell lines but now normalized to i-shCtrl + Doxycycline to determine the effect of MDMX depletion in i-shCtrl versus i-shFOXO cells. Significant alterations (*p* < 0.05) in cell survival are indicated with *. (**C**) Indicated MEL202-derived cell lines (CR-Ctrl and CR-p53; i-shCtrl or i-shFOXO) either containing and i-shCtrl or i-shMDMX shRNA construct were treated with doxycycline (D; 10 ng/mL) for 3 days. Relative survival of the cell lines all normalized to Vehicle (V)-treated samples. Significant alterations (*p* < 0.05) in cell survival are indicated with *. (**D**) Relative survival of the indicated cell lines but now normalized to i-shCtrl + Doxycycline to determine the effect of MDMX depletion in i-shCtrl versus i-shFOXO cells. Significant alterations (*p* < 0.05) in cell survival are indicated with *.

**Figure 5 cancers-14-04482-f005:**
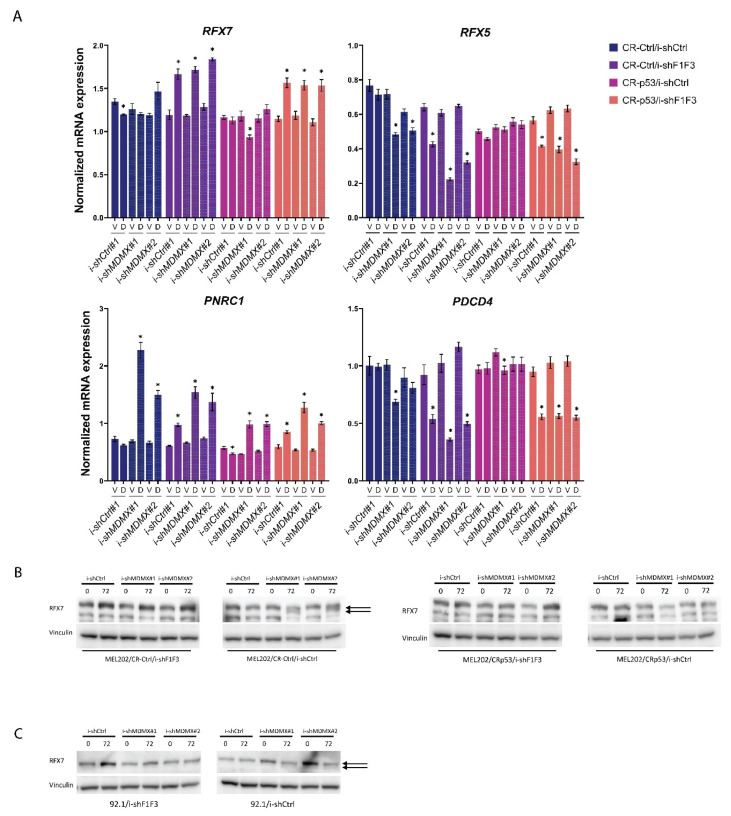
Analyses of RFX7 involvement in transcriptional regulation of genes upregulated upon MDMX depletion. (**A**,**B**) MEL202/CR-Ctrl and/CR-p53, i-shCtrl and i-shF1F3, containing either i-shCtrl or i-shMDMX shRNAs, were treated for 72 h with 10 ng/mL doxycycline (D) or Vehicle (V) after which RNA and protein was harvested and analyzed for the mRNA expression of RFX7 and the reported RFX7-target genes RFX5, PNRC1 and PDCD4 (**A**) and to evaluate the effect of MDMX depletion on RFX7 protein migration. Significant alterations (*p* < 0.05) in gene expression are indicated with *. These are the same samples as analyzed in Figure 3. (**C**) 92.1/i-shCtrl and 92.1/i-shF1F3, i-shCtrl and i-shMDMX cell lines were treated for 72 h with 10 ng/mL doxycycline, after which protein lysates were harvested and analyzed for RFX7 expression. These are the same samples as analyzed in Appendix A. Original blots see Appendix A.

## Data Availability

The RNAseq data sets at GEO Datasets (NCBI) can be found here: https://www.ncbi.nlm.nih.gov/geo/query/acc.cgi?acc=GSE211802 (accessed on 25 August 2022).

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
