# Peer review of "MDMX Regulates Transcriptional Activity of p53 and FOXO Proteins to Stimulate Proliferation of Melanoma Cells"

_cancers, 2022, doi:10.3390/cancers14184482_

Round 1

Reviewer 1 Report

MDM4/MDMX is known to function by restraining p53 activity, thereby promoting oncogenic transformation in a considerable subset of cancers. MDMX forms an interesting therapeutic target because these cancers, as they depend on MDMX expression for their survival. Most focus has been on disrupting the interaction of MDMX with p53, thereby liberating and reactivating p53. However, MDMX also possesses p53 independent functions. Heijkants et al set out to investigate the transcriptome upon MDMX depletion, using mainly two uveal melanoma cell lines known to depend on high levels of MDMX. They find that many genes whose expression is changed upon MDMX knockdown are targets of Foxo transcription factors. Subsequent experimentation further suggest that growth inhibition upon MDMX knockdown is in part mediated by FOXO proteins. This study yields a valuable resource for future studies of MDMX and presents interesting new findings that increase the understanding of MDMX and could be of clinical relevance. In general, this is a well written manuscript and the experiments are clean and well controlled. I have a few comments and recommendations.

1 As mentioned in the discussion, current efforts to pharmacologically target MDMX are mostly aimed at interfering with the interaction with p53, which may not be effective to block p53 independent effects of MDMX. This hypothesis could be tested experimentally, by comparing the effect of MDMX depletion with the effect of pharmacological MDMX inhibitors on Foxo target gene expression and viability in cells with and without p53 and Foxo

2 To what extent is MDM2 also involved in regulating Foxo activity? It should be possible to test this by assessing transcription of a few selected genes upon overexpression of either MDMX or MDM2.

3 Since Foxo depletion alone leads to reduced growth (Fig. 4A and 4C), it is difficult to assess whether the growth inhibition resulting from MDMX knockdown is (in part) due its role in regulating transcription of FOXO target genes. The authors try to dissect these effects by calculating relative survival (Fig. 4B and 4D), indeed showing that the effect of MDMX knockdown is reduced in the FOXO depleted situation. This is a key point of the paper and to better substantiate this, proper statistics are required.

4 Figure 2: V and D (vehicle and doxycycline) are not explained in the legend

5 Supplementary Figure 5; V,D and N (vehicle, doxycycline and Nutlin) are not explained in the legend.

6 The text in many figures is too small (figure 3B, 4A, 4B, 4D, 5A, S3, S5, S6B, S6C)

Reviewer 2 Report

The manuscript entitled “MDMX regulates transcriptional activity of p53 and FOXO proteins to stimulate proliferation of melanoma cells” by Renier C. Heijkants et al., was presented as an original Article in which the authors analyze the role of MDMX protein in the regulation of melanoma cancer cell proliferation, through the inhibition of p53 and FOXO transcription factors. The introduction needs to be revised, adding novelty about this study. Although the conclusions are in line with the presented data, the results could be improved including several experimental approaches and different methodologies. While the paper would be of interest for a wide community working on cancer treatment and therapy, the manuscript is still preliminary, and the quality of the data does not reach the expected quality standards required for publication. Several major points need to be addressed in order to strengthen the technical rigor of the manuscript:

Major comments for the text:                

The manuscript needs to be substantially improved in terms of data presentation and writing to make the work more accessible to the reader, since the article is too long and appears dispersive. There are many issues regarding the presentation of the data in the figures. For example, not all figures are properly cited in the Results section and other are missing (e.g., the Figure 2D is not cited in the Results; Page 10, Line 323: The authors cited Suppl. Figure 3A, instead they referred to Suppl. Figure 3B) and some figure legends do not correspond to the relative Figures (e.g., in the Figure Legend of Supplementary Figure 1 it is necessary correct the correspondence between Figures and Figure Legends; Supplementary Figure Legend 4A does not reflect what is shown in the respective Supplementary Figure 4A). Experimental procedures are not sufficiently described: e.g., how the authors evaluated the cell survival? Concerning the gene nomenclature, where the authors referred to p53 as gene, please replace it with own gene name, i.e. TP53. Moreover, gene name abbreviations (i.e. TP53MDM2MDMX) must be italicized. 

Major comments for the data:

In general, the data presented in the figures need to be improved. For example: 

I)       The reviewer retains that the criteria chosen for the RNA-seq analysis are not correct. Usually, in a differential gene expression analysis is necessary to adopt more stringent criteria. A safe |Log2 Fold Change| (|Log2 FC|) ≥ 1.5 or more is normally adopted as ideal cutoff, with a False Discovery Rate (FDR) ≥ 0.05. Maybe the authors chosen |Log2 FC| ≥ 0.7 because the Differentially Expressed Genes (DEGs) are too few. For the reasons set out above, it would be appropriate to repeat the RNA-seq analysis. To allow a reanalysis of the presented data, RNA-seq dataset must be deposited in a publicly accessible database (i.e. Geo DataSets from NCBI), providing the accession codes and the associated hyperlinks in the Data Availability section. Furthermore, to visualize the differential expression of genes across the samples, the reviewer suggests to show a) a Volcano Plot of all DEGs and b) a hierarchical clustering heatmap of the top DEGs from the RNA-Seq experiment.

II)      GO terms pathway analysis needs to be better represented. The reviewer suggests to represent the data of enrichment analysis for biological process available in Table S4 as bar chart expressing biological processes as a function of gene counts. The authors consider the possibility of using also other tools for the GO enrichment analysis, i.e. Enrichr (https://maayanlab.cloud/Enrichr/), Metascape (https://metascape.org) or GSEA-MSigDB (https://gsea-msigdb.org/gsea/msigdb).

III)    The authors evaluated the effects of MDMX depletion also in 92.1 cells. a) Why do they not show the effects of MDMX depletion on the downregulated genes MAD2L1 and CENPF also in 92.1 cell line? b) Why do they show only two clones in the real-time qPCR analyses and extend the proteins analysis also to a third clone?

IV) Why did the authors choose different times of exposure in Nutlin-3 treatment experiments? (i.e. 24h in MEL202 cells, see Suppl. Figure 4A and 72h in MEL 93.05 cells, see Suppl. Figure 5A). Similarly, why did they choose different times of doxycycline exposure in the experiments? (i.e. 48h in the experiments in Figure 2 and 72h in the experiments in Figure 3 and Supplementary Figure 5?)

V)    Statistic is not shown in any of the figures. It needs to be included. Please, add statistical analysis for each experiment. If no significant differences are proposed, “n.s.” should be indicated. Furthermore, it is necessary improve the Materials and Methods section adding a paragraph relative to Statistic.

VI)  No quantification of Western Blot analyses is shown. Please, provide quantification histograms (ratio of target protein levels relative to USP7 levels) of all Western Blot analyses, both in Main and Supplementary Figures, to determine changes in target proteins expression.

VII)  The reviewer suggests to create a table summarizing the number of all cell clones and the characteristics of all cell lines used (i.e. cell name, cell type, disease, TP53 status, MDMX status).

Major comments for the figures:

The reviewer suggests to reorganize the main and supplementary figures, in order that any figure can be cited in correct order (e.g., Supplementary Figure 4 is cited before Supplementary Figure 3C; Supplementary Figure 6 is cited before Supplementary Figure 5C; Supplementary Figure 7 is cited before Supplementary Figure 6C).

Other comments:

I)  Figure 1A: There is a typo in i-sMDMX#1 and i-sMDMX#3. An “h” is missing in the word sh. Please, correct. 

II) Supplementary Figure 2B: There is a typo in i-shMx#1, i-shMx2 and i-shMx#3. Please, correct Mx with MDMX.

III) Correct the Supplementary Figure Legend 5, citing the Nutlin-3 treatment.

Reviewer 3 Report

Heijkants, et al. provided significant insight of MDMX and its role in tumorigenesis. They showed, induced expression of MDMX protein, stimulates the proliferation of cancer cells by inhibiting the activity of p53 and FOXO transcription factor. Author should describe MDMX roles using other cancer. Overall, manuscript is well written and can be improved further. I have few concerns:

·       What is the rationale for using an only UM cell line?

·       Did the authors validated shRNA hits for MDMX validated independently to reduce MDMX protein levels and cell growth using other MDMX amplified lines? This is crucial as off-target effects of shRNAs are well-documented.

·       It is very difficult to see increase of p53 and p21 proteins. It is recommended quantitate all blots with p53 expression. Also, shCtrl shows more expression of p53 and p21 at 72 hours’ time point (Figure 1A), explain?

·       It is important to do all experiments in triplicates. Author needs to show original western blot results from each experiment.

·       I would be nice if authors show the same phenomena by over-expressing MDMX in cell lines with low expression of MDMX.

·       p-values are missing in all bar graphs. This needs to be checked for all the figures. It is unacceptable to present figures without proper statistics.

·       It’s confusing for the readers to track what is V, D, N, DN etc. decipher this in figure legends.

·       Figure 2D is not cited anywhere in the manuscript.

Round 2

Reviewer 2 Report

The reviewer retains that RNA-seq is a key element of the entire article, but the criteria chosen for the RNA-seq analysis are not acceptable. For the reasons set above, I don't feel to recommend the publication of this article.

Reviewer 3 Report

The authors have satisfactorily addressed most of my concerns. In particular, the authors have greatly streamlined the manuscript by adding statistics and necessary writing in the figures and text. The revised manuscript is ready for publication.